# Adaptive Laboratory Evolution of *Bacillus subtilis* 168 for Efficient Production of Surfactin Using NH$_4$Cl as a Nitrogen Source

Jie Li [1], Weiyi Tao [2], Shenghui Yue [3], Zhangzhong Yuan [3] and Shuang Li [1,*]

[1] College of Biotechnology and Pharmaceutical Engineering, Nanjing Tech University, Nanjing 211816, China; 202061218222@njtech.edu.cn

[2] College of Food Science and Light Industry, Nanjing Tech University, Nanjing 211816, China; taoweiyi@njtech.edu.cn

[3] Research Institute of Petroleum Engineering and Technology, Sinopec Shengli Oilfield, Dongying 257000, China; yueshenghui.slyt@sinopec.com (S.Y.); yuanzhangzhong.slyt@sinopec.com (Z.Y.)

[*] Correspondence: lishuang@njtech.edu.cn; Tel./Fax: +86-25-58139942

**Abstract:** *Bacillus subtilis* strain 168 is commonly used as a host to produce recombinant proteins and as a chassis for bio-based chemicals production. However, its preferred nitrogen source is organic nitrogen, which greatly increases production costs. In this study, adaptive laboratory evolution (ALE) was used to improve *B. subtilis* 168 growth using NH$_4$Cl as the sole nitrogen source. The cell density (OD$_{600}$) of a mutant strain LJ-3 was 208.7% higher than that of the original strain. We also optimized the metal ions in the medium and this resulted in a further increase in growth rate by 151.3%. Reintroduction of the *sfp+* gene into strain LJ-3 led to the LJ-31 clone, which restored LJ-3's ability to synthesize surfactin. The fermentation system was optimized (C/N, aeration, pH) in a 5 L bioreactor. Dry cell weight of 7.4 g/L and surfactin concentration of 4.1 g/L were achieved using the optimized mineral salt medium after 22 h of batch fermentation with a Y$_{P/S}$ value of 0.082 g/g and a Y$_{P/X}$ of 0.55 g/g. HPLC analysis identified the surfactin isoforms produced by strain LJ-31 in the synthetic medium as C$_{13}$-surfactin 13.3%, C$_{14}$-surfactin 44.02%, and C$_{15}$-surfactin 32.79%. Hence, the variant LJ-3 isolated by ALE is a promising engineering chassis for efficient and cost-effective production of a variety of metabolites.

**Keywords:** adaptive laboratory evolution (ALE); inorganic nitrogen; *B. subtilis* 168; surfactin; genetic engineering

## 1. Introduction

In a vast majority of industry-based microbial metabolite production, carbon and nitrogen sources utilized by the microbes to produce raw materials often account for 50–90% of the production costs. Nitrogen sources are either organic or inorganic sources with significant price differences. Organic nitrogen sources are mainly tryptone and yeast extract, which typically cost between USD1800 and 5000/ton, while inorganic nitrogen sources are mainly ammonium salts and nitrates, with prices ranging between USD70 and 150/t. Ammonium chloride is the cheapest inorganic nitrogen source with costs ranging from USD70 to 80/t. Therefore, the price of organic nitrogen sources is usually 35–70 times higher than that of inorganic nitrogen sources and makes up a large proportion of the production costs for industrial microorganisms. Strains that can use ammonium chloride as a nitrogen source efficiently would provide a significant advantage in terms of lowering costs.

*Bacillus subtilis* 168 is a well-known domesticated strain that exhibits natural genetic competence. This strain has a defined genetic background and is a safe and harmless strain for industrial platforms. As this strain has strong protein expression and secretion

abilities, it is widely used for the biosynthesis of enzymes [1], nucleosides [2], antibiotics [3], surfactants [4], lactic acid and isobutanol [5], N-acetylglucosamine [6], and other bio-based chemicals. However, *B. subtilis* strain 168 has a major disadvantage in that it prefers organic nitrogen (tryptone and yeast extract), which greatly increases production costs. Strains of *B. subtilis* that can utilize inorganic nitrogen are usually wild-type or parental strains, which are generally not amenable to gene editing. Therefore, the availability of *B. subtilis* 168, derived from *B. subtilis* Marburg, that can effectively utilize inorganic nitrogen, is highly desirable. However, not much is known about how *B. subtilis*, which prefers organic nitrogen, can be converted to efficiently utilize inorganic nitrogen. This may be because nitrogen assimilation is a complex system that is regulated by multiple gene products [7], making it difficult to alter the capacity for nitrogen assimilation by editing one or a few genes.

Adaptive laboratory evolution (ALE) is a narrow-range experimental evolutionary concept that artificially applies stress to stimulate the natural environment and obtain a desired phenotype. ALE was originally used by Dallinger in a 7-year high-temperature adaptation experiment [8]. Since then, ALE was widely used as an efficient tool for biological discovery and industrial biotechnology, such as substrate utilization, growth rate optimization, increasing tolerance, and increasing product yield/titer [9]. In the present study, a variant of *B. subtilis* 168 that can effectively utilize $NH_4Cl$ as a nitrogen source was isolated by ALE. Then, trace metal elements in the synthesis medium were optimized to further increase the biomass of the bacteria. Subsequently, the *B. subtilis* 168 *sfp+* variant was created to produce surfactin in the optimized mineral salt medium. At the same time, the composition of the product was further analyzed, as shown in Figure 1.

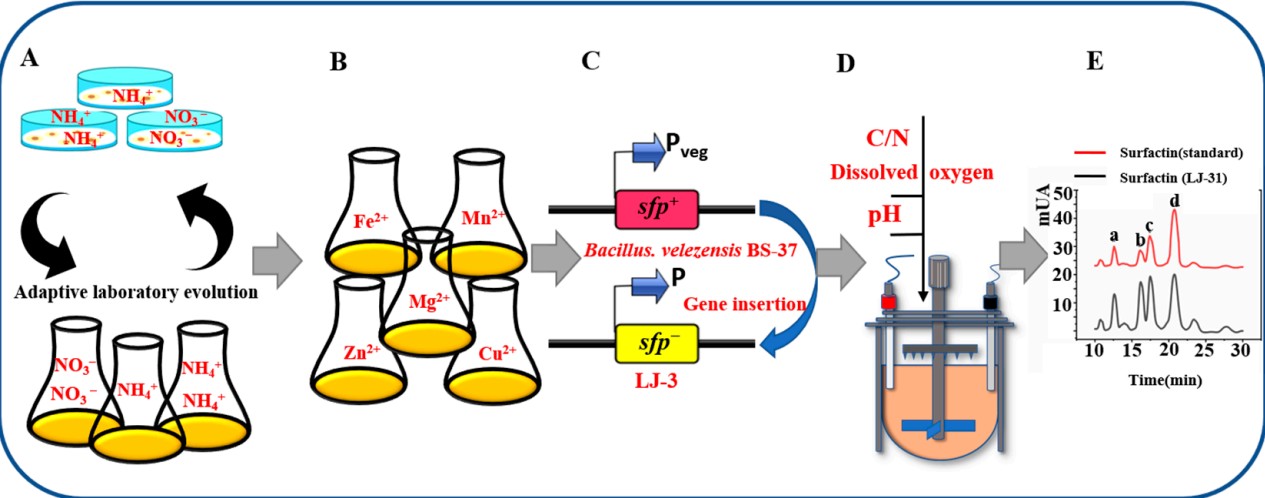

**Figure 1.** Experimental design process diagram. (**A**) Isolation of a *B. subtilis* 168 mutant that efficiently utilizes inorganic nitrogen. (**B**) Optimization of trace metals in the culture medium. (**C**) The *sfp+* gene from *B. velezensis* BS-37 was transformed into LJ-3 to restore the ability of LJ-3 to synthesize surfactin. (**D**) Fermentation regulation. (**E**) Analysis of surfactin composition: a—$C_{13}$-surfactin, b—iso-$C_{14}$-surfactin, c—n$C_{14}$-surfactin, and d—anteiso-$C_{15}$-surfactin.

## 2. Materials and Methods

### 2.1. Strains and Primers

The strains, plasmids, and primers used in this study are listed in Tables 1 and 2.

**Table 1.** Strains and plasmids used in this study.

| Names | Characteristics | References |
|---|---|---|
| Strains | | |
| *B. subtilis* 168 | start strain | This paper |
| *B. subtilis* 168 (P$_{veg}$-GFP) | *B. subtilis* 168 derivative, containing P$_{veg}$ promoter (Strong promoter of *Bacillus subtilis*) | [10] |
| *Bacillus. velezensis* BS-37 | Wild-type, surfactin producer | [11] |
| LJ-3 | *B. subtilis* 168 variant | This study |
| LJ-31 | LJ-3 derivative, *sfp+*, Overexpression of the *sfp* gene derived from *B. velezensis* BS-37 was regulated by the promoter P$_{veg}$ | This study |
| Plasmid | | |
| pNZT1-pheS* | pNZT1 with P$_{bc}$-*pheS*-cat* cassette | [12] |

**Table 2.** Primers used in this study.

| Primers | Primer Sequence (5′-3′) | Gene Fragment |
|---|---|---|
| LF-F | TTTGTGATTTTCAGCGTGATTGAAAACCT | LF |
| LF-R | CTGTGTAAGATAGATCTCTAGATCCTCCGTCTGCAAAAGATTGT | |
| Pveg-F | AGACGGAGGATCTAGAGATCTATCTTACACAGCATCACACTGG | P$_{veg}$ |
| Pveg-R | ACTCCGTAAATCTTCATGTTTGTCCTCCTTATTAGTTAATCTACATTTAT | |
| Sfp-F | AATAAGGAGGACAAACATGAAGATTTACGGAGTATATATGGACCGC | *Sfp+* |
| Sfp-R | GCGCACTGAAAAGGAATTATAACAGCTCTTCATACGTTTTCATCTCAATC | |
| DR-F | GAGATGAAAACGTATGAAGAGCTGTTATAATTCCTTTTCAGTGCGCCTGC | DR |
| DR-R | TCATTTGTATACATACTTTAAAAATAGATTATCCGAAAGAAAATCTATTA | |
| PC-F | TAATAGATTTTCTTTCGGATAATCTATTTTTAAAGTATGTATACAAATGA | PC |
| PC-R | ATAAATTCCGTAAATCTTCATTTATAAAAGCCAGTCATTAGGCCTATCTG | |
| RF-F | CCTAATGACTGGCTTTTATAAATGAAGATTTACGGAATTTATATGGACCG | RF |
| RF-R | TCTCCTTGAGGCGATAGACCG | |

Underlined letters represent homologous sequences for fusion PCR.

*2.2. Culture Media*

The seed medium contained (per L): 10 g tryptone, 5 g yeast extract, and 10 g sodium chloride. ALE medium contained (per L): 20 g sucrose, 50 mM inorganic nitrogen (NaNO$_3$ or NH$_4$Cl), 3 g KH$_2$PO$_4$, 10 g K$_2$HPO$_4$·3H$_2$O, 0.5 g MgSO$_4$·7H$_2$O, and 0.05 g FeSO$_4$·7H$_2$O. Agar (20 g/L) was added to make solid medium. Basic medium components for metal ion optimization contained (per L): 20 g sucrose, 50 mM NH$_4$Cl, 3 g KH$_2$PO$_4$, and 10 g K$_2$HPO$_4$·3H$_2$O; to optimize the metal ions in the synthetic medium, metal ions of different concentrations (mM) were added separately to the base medium: Fe$^{2+}$ (0, 0.25, 0.5, 0.75, 1, and 1.25), Mn$^{2+}$ (0, 0.002, 0.01, 0.05, 0.1, and 0.5), Mg$^{2+}$ (0, 0.5, 1, 2, and 4), Cu$^{2+}$ (0, 0.05, 0.1, 0.2, and 0.4), Zn$^{2+}$ (0, 0.05, 0.1, 0.2, and 0.4). The fermentation medium for surfactin in a flask contained (per L): 50 g sucrose, 50 mM NH$_4$Cl, 3 g KH$_2$PO$_4$, 10 g K$_2$HPO$_4$·3H$_2$O, 0.5 mM FeSO$_4$·7H$_2$O, 0.01 mM MnSO$_4$.H$_2$O, and 2 mM MgSO$_4$·7H$_2$O. The fermentation medium for surfactin in a 5 L bioreactor contained (per L): 50 g sucrose, 50–250 mM NH$_4$Cl, 3 g K$_2$HPO$_4$·3H$_2$O, 0.5 mM FeSO$_4$·7H$_2$O, 0.01 mM MnSO$_4$.H$_2$O, 2 mM MgSO$_4$·7H$_2$O, and 1 g of the antifoam agent organic silicone oil. Minimum glycerol medium supplemented with 5 mM p-Cl-Phe (MGY-Cl medium) was the same as that used by Hu [13] et al. Chloramphenicol (5 μg/mL) was used to select transformants when necessary.

### 2.3. ALE of B. subtilis 168 and Optimization of Metal Ions in Synthetic Medium

*B. subtilis* 168 variants that use inorganic nitrogen sources efficiently were isolated by ALE. *B. subtilis* 168 stocks were streaked onto ALE medium plates and incubated for 24 h at 37 °C. Then, several single colonies were picked from the plates, inoculated into 50 mL liquid ALE medium in shake flasks, and incubated for 24 h at 37 °C in a shaker at 200 r/min. After 24 h, the cell density ($OD_{600}$) was measured, and high cell density cultures were diluted and incubated for 24 h at 37 °C on solid ALE medium plates. The variants were selected by passaging through solid and liquid media over several generations.

To optimize the metal ions in the synthetic medium, metal ions ($Fe^{2+}$, $Mn^{2+}$, $Mg^{2+}$, $Cu^{2+}$, and $Zn^{2+}$) of various concentrations (mM) were added separately to the basic medium. The *B. subtilis* 168 mutant strain (LJ-3) needed to be pre-cultured once. Single colonies were picked from the plates, transferred to 250 mL shake flasks supplemented with 50 mL of seed medium, and incubated for 12 h at 37 °C in a shaker at 200 r/min. Then, the seed culture was transferred to 50 mL basic medium at a 4% (*v/v*) inoculum and incubated at 37 °C in a shaker at 200 r/min. Cell density and substrate consumption during 30 h of cultivation were measured.

### 2.4. sfp+ Strain of B. subtilis 168 Variant (LJ-31)

The gene editing approach taken to perform marker-free knock-in gene replacement (Figure 2) was described in detail in our previous work [12,13]. *B. subtilis* containing the mutant alpha subunit of the phenylalanyl-tRNA synthetase gene (*pheS\**) will not survive in the presence of 4-chloro-DL-phenylalanine. The mutant gene (*pheS\**) was inserted within a counter-selectable cassette (containing a chloramphenicol resistance gene (*cat*) and *pheS\** gene, and this cassette is referred to as PC). The strains and plasmids are listed in Table 1 and primers listed in Table 2 were used to amplify the following gene fragments: left flanking of *sfp⁻* gene (LJ-3, LF) (~1000 bp), $P_{veg}$ strong promoter (*B. subtilis* 168, $P_{veg}$-GFP), insertion gene (*B. velezensis* BS-37, *sfp+*), direct repeat (LJ-3, DR) (~400 bp) sequence, counter-selectable cassette (PC), and right flanking of LF (LJ-3,RF) (~800 bp). These fragments were then fused by overlapping PCR in the order LF-$P_{veg}$-*sfp+*-DR-PC-RF. The PCR products were introduced into the LJ-3 and transformants were selected on chloramphenicol and MGY-Cl containing agar plates. The positive transformants were named LJ-31.

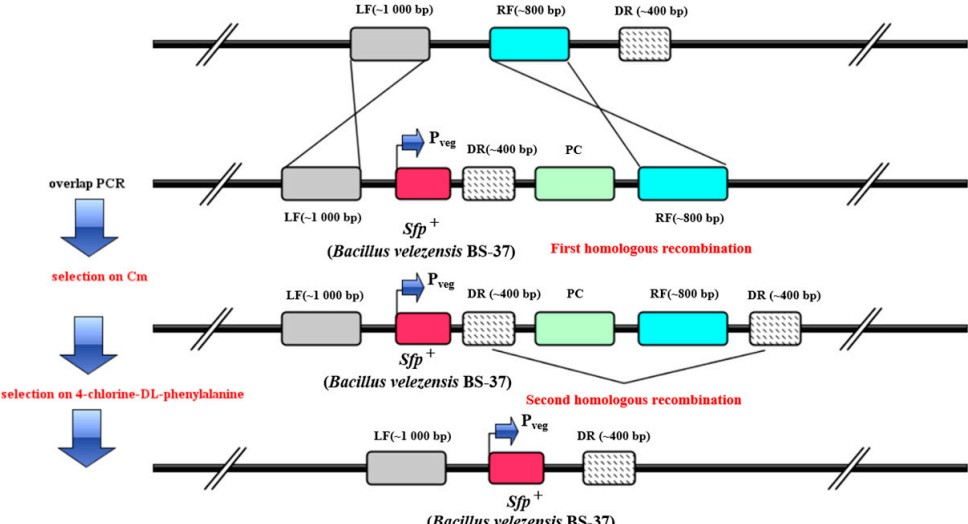

**Figure 2.** The *sfp⁺* version of variant. The *sfp⁻* gene of LJ-3 was replaced with the *sfp⁺* gene from *B. velezensis* BS-37. LF—*sfp⁻* (LJ-3) gene left flanking (~1000 bp); DR—*sfp⁻* (LJ-3) gene right flanking (~400 bp); $P_{veg}$—strong promoter; *sfp⁺* gene (*B. velezensis* BS-37); RF—right flanking of LF (~800 bp); and PC—counter-selectable cassette (containing chloramphenicol resistance gene (*cat*) and *pheS\** gene, named PC).

### 2.5. Surfactin Production in Shake Flasks

The LJ-31 strains from a frozen storage tube were picked out, streaked onto the solid seed medium plate, and incubated for 12 h at 37 °C. Single colonies were picked from the plates and inoculated into shake flasks containing 50 mL of seed medium and incubated for 12 h at 37 °C in a shaker at 200 r/min. The fresh cells were transferred to 50 mL of optimized synthetic medium at a 4% (*v/v*) inoculum and incubated at 37 °C in a shaker at 200 r/min. Cell density, pH, and surfactin concentrations were measured.

### 2.6. Batch Fermentation for Surfactin Production in a 5L Bioreactor

A pre-culture of strain LJ-31 was set up where cells were inoculated into 50 mL of seed medium, and incubated at 37 °C in a shaker at 200 r/min until the cell density reached $OD_{600}$ of 3–4. The culture was then inoculated into the bioreactor at a 4% (*v/v*) inoculum. The working volume of the 5 L bioreactor was 3 L and industrial oxygen was used to replace air. The aeration rate was set to 200 mL/min and the stirring speed was 300 rpm. The pH was regulated with 1 M NaOH and 1 M HCl. Organic silicone oil was used as an antifoam agent at a concentration of 1/1000 (*w/v*).

### 2.7. Determination of Surfactin Concentration

Surfactin concentration was determined using the Shimadzu LC-20 (HPLC) SPD-20A detector equipped with a Venus ll XBP C18 column (4.6 × 150 mm, 5 µm) [13]. The mobile phase consisted of 90% (*v/v*) methanol and 10% (*v/v*) water, with 0.05% (*v/v*) formic acid at a flow rate of 0.6 mL/min. The sample size was 20 µL and the absorbance of the eluent was monitored at 214 nm. Pure surfactin (98%) was purchased from Sigma Aldrich (St. Louis, MO, USA).

### 2.8. Analysis

Sucrose concentration was determined using the Shimadzu LC-20 (HPLC) RID detector equipped with an Aminex HPX-87H, 300 mm × 7.8 mm column (250 mm × 4.6 mm, 5 µm) [14]. The mobile phase was 10 mM sulfuric acid, a column temperature of 30 °C, and a 10 µL injection volume at a flow rate of 0.5 mL/min. $NH_4^+$ concentration was determined using a test kit (Quanzhou Ruixin Biotechnology Co., Ltd., Quanzhou, China; item number: RXWB0375-96). Multiple 10 mL cell suspensions were centrifuged and washed with distilled water. The dry cell weight (DCW) of the cell biomass was determined by drying the cell pellet for 24 h at 70 °C to a constant weight.

### 2.9. Calculation of Dynamic Parameters

The main parameters were sucrose concentration (S, g/L), biomass DCW (X, g/L), and surfactin concentration (P, g/L). The product per substrate ($Y_{P/S}$) and product per biomass ($Y_{P/X}$) were determined at the maximum surfactant concentration using the Equations (1) and (2) [15]. The specific cell growth rate ($\mu$, $h^{-1}$) and specific surfactin production rate ($q_P$, $h^{-1}$) were estimated using Equations (3) and (4) based on the experimental or fitted data of cell growth ($\chi$, g/L) [16].

$$Y_{P/S} = \frac{p}{\Delta S} \Big|_{P=Pmax} \tag{1}$$

$$Y_{P/X} = \frac{p}{X} \Big|_{P=Pmax} \tag{2}$$

$$\mu = \frac{1}{x}\frac{dx}{dt} = \frac{1}{x}\lim_{\Delta t \to 0}\frac{\Delta x}{\Delta t} \tag{3}$$

$$q_P = \frac{1}{x}\frac{dp}{dt} = \frac{1}{x}\lim_{\Delta t \to 0}\frac{\Delta p}{\Delta t} \tag{4}$$

Calculation of theoretical yield:

A surfactin molecule contains four leucines, one glutamate, one valine, and one aspartate as well as a long-chain fatty acid. The surfactin synthesized by the LJ-31 strain

has three major fatty acid chain lengths: $C_{13}$-surfactin, $C_{14}$-surfactin, and $C_{15}$-surfactin. The average fatty acid length is 14 carbon atoms, and a fatty acid chain containing 14 carbon atoms was the most abundant. Therefore, we normalized the fatty acid chain length to 14 carbon atoms [17]. Thus, surfactin contained about 52 carbons with a relative molecular weight of $M_{sur}$ 1022 [10]. About 5.75 sucrose molecules are required to synthesize one molecule of surfactin.

The theoretical yield: 0.174 mol surfactin/mol sucrose.

## 3. Results and Discussion

### 3.1. ALE of B. subtilis 168

Ammonium salts and nitrates are two of the cheapest nitrogen sources, and *B. subtilis* is a safe, non-toxic strain with strong protein expression and secretion properties. As a cell factory, *B. subtilis* 168, which can efficiently utilize inorganic nitrogen, is an attractive choice for a host cell to produce functional products. In the present study, random mutants of *B. subtilis* 168 that efficiently utilized inorganic nitrogen were identified by ALE (Figure 1A). After twenty cycles of adaptive evolution of the parental strain in medium containing $NH_4^+$ or $NO_3^-$, two variants were selected and their capacity for using $NH_4^+$ or $NO_3^-$ was enhanced as expected (Figure 3). The two adapted strains were designated as LJ-2 and LJ-3, respectively. The LJ-2 strain was isolated in ALE medium containing $NO_3^-$, and the LJ-3 strain was isolated in ALE medium containing $NH_4^+$.

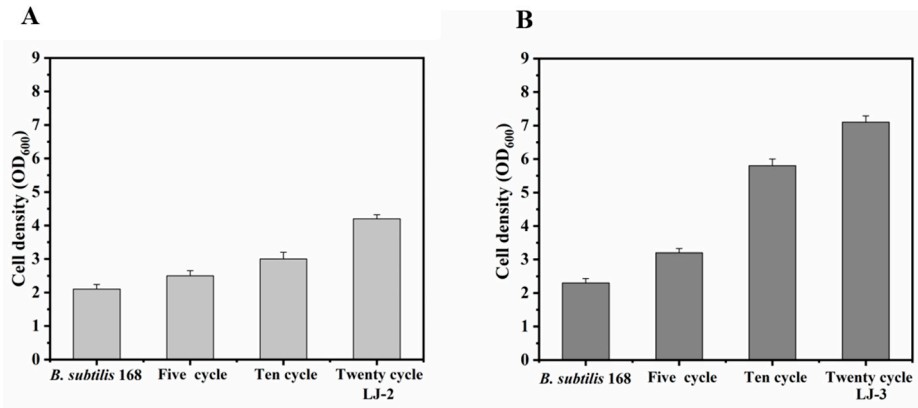

**Figure 3.** ALE of *B. subtilis* 168 incubated in medium with $NO_3^-$ or $NH_4^+$. (**A**) The adaptive evolution of LJ-2 strain in ALE medium with $NO_3^-$ (solid to liquid for one cycle). (**B**) The adaptive evolution of LJ-3 strain in ALE medium with $NH_4^+$ (solid to liquid for one cycle).

As shown in Figure 3, cell density ($OD_{600}$) of the two variants was measured after five cycles, ten cycles, and twenty cycles (one cycle represents one shift from solid to liquid medium). After twenty cycles of passaging, LJ-2 exhibited an increase in cell density from $2.1 \pm 0.14$ to $4.2 \pm 0.12$ (Figure 3A), whereas LJ-3 cell density was more significantly increased from $2.3 \pm 0.13$ to $7.1 \pm 0.19$ (Figure 3B). The cell density of LJ-2 and LJ-3 had an overall increase of 100% and 208.7%, respectively, in comparison to their starter strain. It is worth noting that the presence of $NH_4^+$ provided a better evolutionary adaptive effect compared to $NO_3^-$. This may be because ammonium is an important precursor of glutamine and glutamate and is more easily available to the cells via diffusion or by the ammonium transporter (NrgA) [18]. Meanwhile, the poor adaptive evolutionary effect of nitrate may be related to host strain characteristics, ALE medium, or strain generation. Therefore, further studies are needed to isolate *B. subtilis* 168 that efficiently use nitrate. Overall, the ALE technique is a viable method to improve nitrogen assimilation by *B. subtilis* 168. The LJ-3 strain was next used as the engineering chassis to introduce the surfactin-encoding gene.

### 3.2. Effect of Metal Ions on Growth of LJ-3

Cell density and the yield of functional products usually show a positive linear correlation, and therefore, high cell density results in a high yield of functional products [19]. However, production strains that use inorganic nitrogen usually exhibit lower cell density [20]. Trace metal elements are essential for bacterial growth. Metal ions were also essential for bacteria to promote the uptake of $NH_4^+$ or $NO_3^-$ [20–22]. To further promote LJ-3 cell growth, the effects of five metal ions ($Fe^{2+}$, $Mn^{2+}$, $Mg^{2+}$, $Cu^{2+}$, and $Zn^{2+}$) on cell growth and the utilization of different substrates were examined (Figure 4).

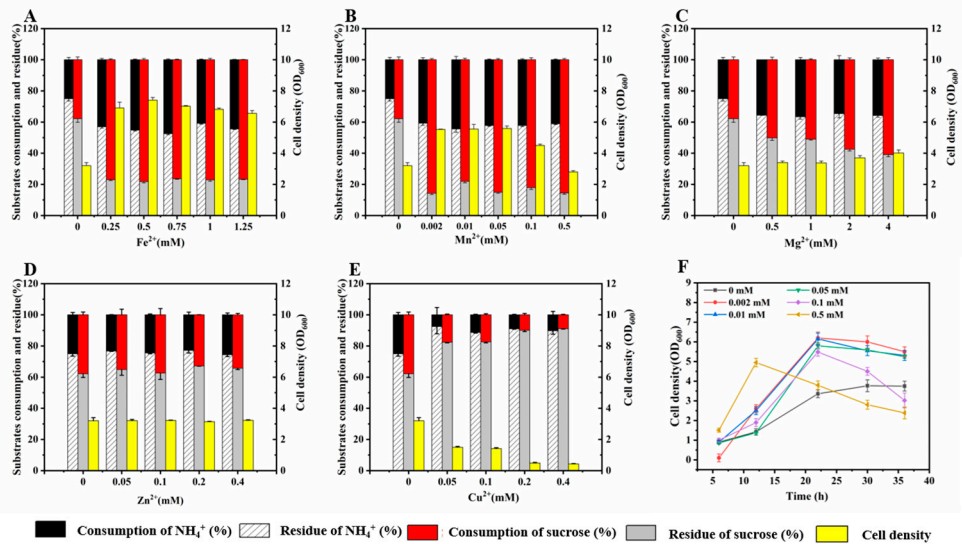

**Figure 4.** Effects of different metal ions on LJ-3 substrate utilization and cell growth. (**A**) Effect of different concentrations of $Fe^{2+}$. (**B**) Effect of different concentrations of $Mn^{2+}$. (**C**) Effect of different concentrations of $Mg^{2+}$. (**D**) Effects of different concentrations of $Zn^{2+}$. (**E**) Effects of different concentrations of $Cu^{2+}$. (**F**) Cell density varies with time under different concentrations of $Mn^{2+}$. Consumption of $NH_4^+$ (%) (black squares), residual $NH_4^+$ (%) (white squares), consumption of sucrose (%) (red squares), residual sucrose (%) (grey squares), and cell density (yellow squares) were measured.

As shown in Figure 4A, $Fe^{2+}$ could promote the utilization of $NH_4^+$ and sucrose as well as cell growth. $Fe^{2+}$ at 0.05 mM was optimal for LJ-3, and utilization of $NH_4^+$ and sucrose after 30 h of culture was 45.2% and 78.16%, respectively. These values were 80.8% and 106.8% higher relative to the control (0 mM $Fe^{2+}$). In parallel, cell density achieved a reading of 7.41 ($OD_{600}$), which was an increase of 131.5%. $Mn^{2+}$ also promoted the utilization of substrates and cell growth (Figure 4B). Interestingly, we noted a significant difference in cell densities over different concentrations of $Mn^{2+}$ at 30 h. Cell density was high in the presence of low concentrations of $Mn^{2+}$ (0.002, 0.01, and 0.05 mM), but low cell density was observed with higher concentrations of $Mn^{2+}$ (0.1, 0.5 mM). We also observed that higher concentrations of $Mn^{2+}$ tended to accelerate cell senescence during cell growth (Figure 4F), similar to that previously reported by Mei [22] et al. Therefore, a concentration of 0.01 mM $Mn^{2+}$ was selected to promote the utilization of substrates and cell growth while ensuring cell viability. The utilization of $NH_4^+$ and sucrose by LJ-3 in the presence of 0.01 mM $Mn^{2+}$ over a period of 30 h was 44.4% and 78%, respectively, which were 77.6% and 106.4% higher relative to the control (0 mM $Mn^{2+}$), and the cell density readout reached 5.5, an increase of 92.2%.

For $Mg^{2+}$, cell density increased linearly relative to the increase in $Mg^{2+}$ concentration, which helped to improve the utilization of substrates and cell growth (Figure 4C). However, the high concentrations of $Mg^{2+}$ tended to precipitate during high-temperature treatment, resulting in the blockage of the gas distributor or difficulty in cleaning the bioreactor. Therefore, 2 mM $Mg^{2+}$ was chosen (Figure 4C) and the utilization of $NH_4^+$ and sucrose

at 30 h by LJ-3 was 34.6% and 57.4%, respectively, which were 38.4% and 51.8% higher relative to the control (0 mM $Mg^{2+}$), while cell density reached 3.7, an increase of 15.6%. However, when different concentrations of $Zn^{2+}$ were added (Figure 4D), utilization of $NH_4^+$ and sucrose as well as cell density did not change significantly compared to the control (0 mM $Zn^{2+}$). On the other hand, when different concentrations of the $Cu^{2+}$ were added, the utilization of $NH_4^+$ and sucrose as well as cell growth was inhibited (Figure 4E). Cell growth was inhibited at lower concentrations of $Cu^{2+}$ (0.05 mM). When compared to the control cells (0 mM $Mn^{2+}$), the utilization of $NH_4^+$ and sucrose was reduced by 70% and 53.7%, respectively, and the cell density was reduced by 53.1%.

Taken together, the results indicate that $Fe^{2+}$, $Mn^{2+}$, and $Mg^{2+}$ could promote the uptake of $NH_4^+$ and sucrose as well as cell growth, but not $Zn^{2+}$ and $Cu^{2+}$. Among them, $Fe^{2+}$ had the greatest effect, followed by $Mn^{2+}$ and finally $Mg^{2+}$. Iron homeostasis in bacteria is maintained by various iron uptake systems, which are indispensable for many key enzymes of the cell, such as ribonucleotide reductases, superoxide dismutases, and various energy-generating and regulatory proteins of heme or iron–sulfur cluster cofactors [23]. $Mn^{2+}$ and $Mg^{2+}$ enhance nitrogen assimilation by increasing glutamine synthetase (GS) and glutamate synthetase (GOGAT) activities [21,24,25].

When single metal ions were added, sucrose consumption was approximately 80%, which may eventually be a limiting factor for cell growth. When sucrose concentration was increased to 50 g/L with the addition of metal ions (0.5 mM $Fe^{2+}$, 0.01 mM $Mn^{2+}$, and 2 mM $Mg^{2+}$), LJ-3 cell density ($OD_{600}$) at 24 h reached 9.36, which was 291.6% higher relative to *B. subtilis* 168 ($OD_{600}$ of 2.3) (Figure 5A). The maximum growth rate of LJ-3 was 0.47 $h^{-1}$ with a lag phase of 7 h and it exhibited a 151.3% increase in growth rate compared with the reference strain (Figure 5B). This result confirmed that when cell growth was limited in media supplemented with $NH_4^+$, adding metal ions could promote substrate uptake and cell growth. The combination of ALE technology and metal ion screening and optimization could effectively solve the problem of nitrogen assimilation by certain bacterial hosts, and isolated mutant bacteria can be utilized to produce a variety of bio-based products at a low cost.

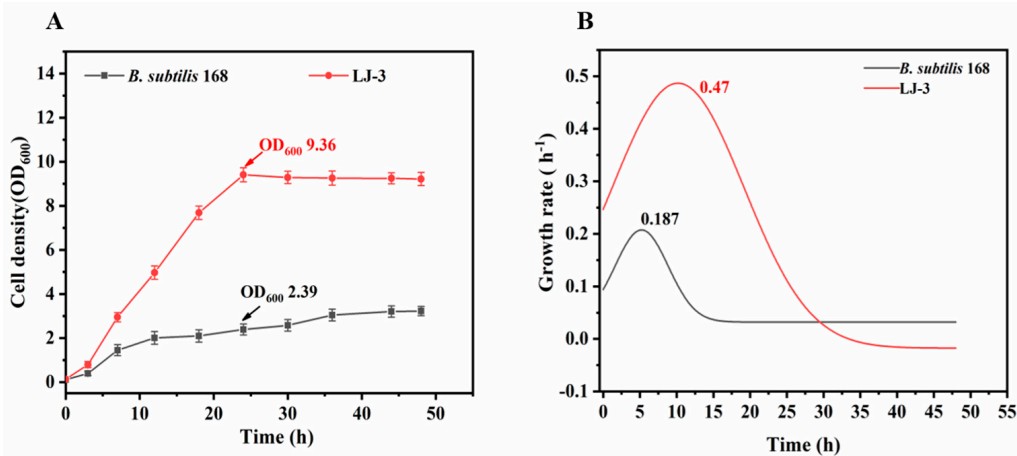

**Figure 5.** Beneficial metal ions (0.5 mM $Fe^{2+}$, 0.01 mM $Mn^{2+}$, and 2 mM $Mg^{2+}$) were added to promote cell growth in the presence of $NH_4^+$ after adaptive laboratory evolution. (**A**) The growth of LJ-3 and *B. subtilis* 168 in optimized medium. (**B**) Growth rates of LJ-3 and *B. subtilis* 168.

### 3.3. LJ-3 Was Used as Engineering Chassis to Produce Surfactin

Sfp protein (4-phosphopantothionyl transferase) is indispensable for the synthesis of surfactin. Sfp transfers the 4′-phosphopantothionyl moiety of coenzyme A to the serine residue of the peptidyl carrier protein (PCP) module to activate the synthesis of surfactin. However, the *sfp* gene sequence in *B. subtilis* 168 contains a termination codon that results in a truncated and inactive Sfp protein [17]. Therefore, in this study, the *sfp+* gene from *B. velezensis* BS-37 [26], a wild-type strain that can produce surfactin efficiently, was

transformed into LJ-3 to replace the mutant *sfp⁻* gene and restore the ability to synthesize surfactin (Figure 2). This transformed strain is henceforth referred to as LJ-31. LJ-31 was incubated in shake flasks and the cell density reached 11.58, which was 23.7% higher relative to LJ-3 (Figure 6A). This confirmed that surfactin synthesis capacity was restored, but the concentration was low at only 0.77 g/L (Figure 6B). In addition, the concentration was depleted to 0.28 g/L at 18–48 h when the pH was maintained between 5 and 6. The culture pH had a significant impact on surfactin production with *Bacillus* strains [27]. Therefore, the most likely reason for the low yield of surfactin is pH. Hence, verification of the effect of pH on surfactin synthesis requires further studies at the bioreactor level.

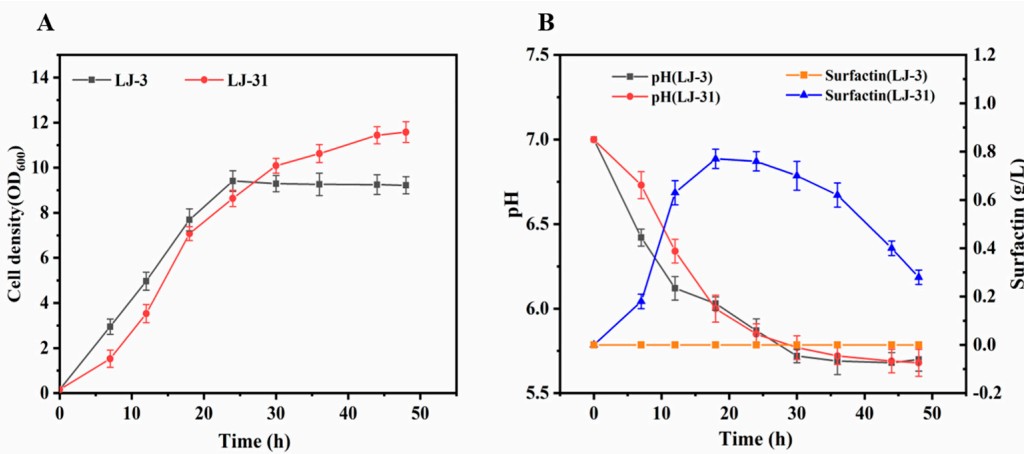

**Figure 6.** The ability of LJ-3 to synthesize surfactant was restored in the LJ-31 transformant, and the cell density, pH, and surfactin yield were monitored. (**A**) Growth curves of LJ-3 and LJ-31. (**B**) The change in pH in medium and accumulation of surfactin.

### 3.4. Production of Surfactin by LJ-31 in a Bioreactor

We developed a non-overfoaming fermentation system for surfactin production in a 5 L bioreactor. Industrial oxygen was used to replace air as the gas supply. The initial parameters were set at a working volume of 3 L, aeration volume of 0.06 vvm (oxygen), stirring speed of 300 rpm, and pH 6.5 and organic silicone oil of 1/1000 (*w/v*) as antifoam agent. The yield of surfactin reached 1.64 g/L at 22 h, which was 113% higher than that in the shake flask (Figure 7A). As shown in Figure 7B, the maximum specific production rate $q_{pmax}$ was 0.2 h$^{-1}$ at 10 h, and the strain reached the maximum specific growth rate $\mu_{max}$ = 0.28 h$^{-1}$ at 7.5 h, indicating that the strain grew rapidly at 7.5 h, and then initiated synthesis at a very rapid rate at 10 h. At 22 h of fermentation, the product per substrate ($Y_{P/S}$) value was 0.055 g/g, and product per biomass ($Y_{P/X}$) was 0.35 g/g. The surfactin yield reached 18.4 mmol/mol sucrose (10.6% of the theoretical yield). The results indicate that surfactin synthesis was sensitive to pH, and needed to be regulated when $NH_4Cl$ was used as a nitrogen source. Therefore, fermentation conditions were further optimized to maximize surfactin production.

Studies showed that C/N, dissolved oxygen, and pH are key factors affecting synthesis of surfactin [27–30]. We further optimized the C/N, dissolved oxygen, and pH to improve the efficiency of surfactin synthesis. The sucrose content was kept constant at 50 g/L, and the C/N ratio was changed by increasing the nitrogen source. Under the conditions of pH 6.5 and aeration volume of 0.06 vvm, the C/N ratio was set at four gradient values (35/1, 17.51/1, 11.68/1, and 7/1). The results are shown in Table 3, and a suitable ratio of C/N for surfactin synthesis is presented. When $NH_4Cl$ was added at 150 mM (C/N 11.68/1), surfactin yields achieved the highest reading of 3.82 g/L, which was 131.5% higher than before optimization (1.65 g/L), and the product per substrate ($Y_{P/S}$) was 0.076 g/g. The dry cell weight (DCW) reached 8.24 g/L, and product per biomass ($Y_{P/X}$) was 0.46 g/g. At this phase, substrate consumption was in a state of carbon source depletion and nitrogen source surplus. However, surfactin amounts decreased by 8.3% (3.5 g/L) and DCW increased by

30.1% (10.72 g/L) when the nitrogen source was increased to 200 mM (C/N 7/1), but the nitrogen source surplus was greater. The excess availability of nitrogen potentially directed the carbon source towards bacterial growth and metabolism. Therefore, the C/N ratio has a significant effect on surfactin synthesis, and an appropriate C/N needs to be regulated.

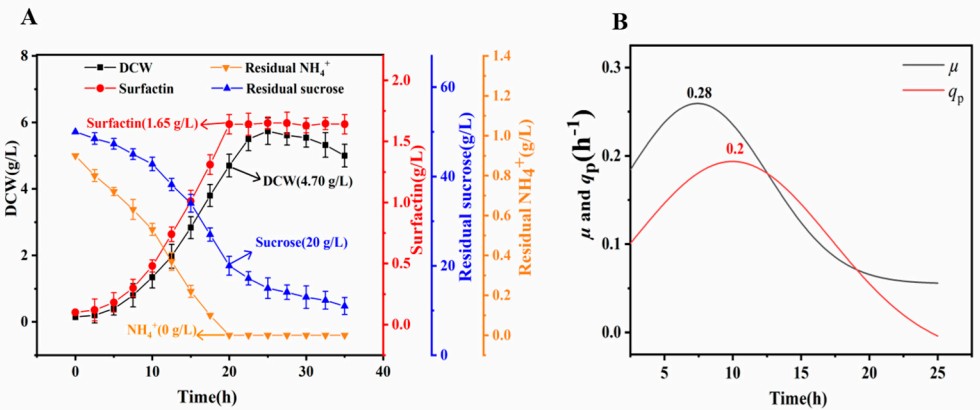

**Figure 7.** Effect of pH on the synthesis of surfactin by LJ-31 using NH$_4$Cl and variation in relevant kinetic parameters ($\mu$, $q_p$). (**A**) Changes in substrate, dry cell weight, and surfactin yield at pH 6.5. (**B**) Changes in specific yield of surfactin ($q_p$) and specific growth rate ($\mu$) of LJ-31.

**Table 3.** The C/N, dissolved oxygen, and pH were optimized in a bioreactor. Substrate, DCW, and surfactin were measured, and the product per substrate and product per biomass were calculated at 22 h.

| Conditions | | Sucrose | NH$_4^+$ | DCW | P$_{max}$ | Y$_{P/S}$ | Y$_{P/X}$ |
|---|---|---|---|---|---|---|---|
| | | (g/L) | (g/L) | (g/L) | (g/L) | (g/g) | (g/g) |
| pH 6.5 Oxygen supply (vvm): 0.06 | | | | | | | |
| | 35 | 20 ± 1.2 | 0 | 4.70 ± 0.20 | 1.65 ± 0.05 | 0.055 | 0.35 |
| C/N | 17.51 | 7.6 ± 2.0 | 0.55 ± 0.18 | 6.62 ± 0.12 | 1.90 ± 0.04 | 0.045 | 0.29 |
| (mol/mol) | 11.68 | 0 | 0.81 ± 0.12 | 8.24 ± 0.23 | 3.82 ± 0.06 | 0.076 | 0.46 |
| | 7 | 0 | 2.30 ± 0.18 | 10.72 ± 0.17 | 3.50 ± 0.07 | 0.070 | 0.33 |
| pH 6.5 C/N (mol/mol): 11.68 | | | | | | | |
| | 0.13 | 6.4 ± 1.5 | 1.29 ± 0.11 | 6.68 ± 0.21 | 1.60 ± 0.08 | 0.037 | 0.24 |
| Oxygen supply (vvm) | 0.06 | 0 | 0.81 ± 0.12 | 8.24 ± 0.33 | 3.82 ± 0.06 | 0.076 | 0.29 |
| | 0.03 | 0 | 0.90 ± 0.10 | 7.40 ± 0.25 | 4.10 ± 0.04 | 0.082 | 0.55 |
| | 0.02 | 6.93 ± 1.3 | 1.27 ± 0.17 | 6.10 ± 0.21 | 2.55 ± 0.05 | 0.059 | 0.42 |
| C/N (mol/mol): 11.68 Oxygen supply (vvm): 0.03 | | | | | | | |
| | 6 | 1.50 ± 1.2 | 0.94 ± 0.12 | 8.11 ± 0.23 | 2.56 ± 0.05 | 0.053 | 0.32 |
| pH | 6.5 | 0 | 0.90 ± 0.10 | 7.40 ± 0.35 | 4.10 ± 0.04 | 0.082 | 0.55 |
| | 7 | 0.17 ± 2.1 | 0.90 ± 0.15 | 8.29 ± 0.32 | 3.85 ± 0.06 | 0.077 | 0.46 |
| | 7.5 | 0.19 ± 1.9 | 0.70 ± 0.18 | 8.97 ± 0.26 | 3.70 ± 0.07 | 0.074 | 0.41 |

At a fixed C/N ratio of 11.68/1 and pH 6.5, the aeration volume was optimized. The results show that aeration volume was also a key factor affecting surfactin synthesis. As shown in Table 3, the optimal aeration volume for surfactin production was 0.03 vvm, whereby the surfactin yield reached 4.10 g/L, which was 7.3% higher than that of the control (3.82 g/L) and the DCW was 7.4 g/L, which was a slight decrease of 10.2%, but the residual nitrogen source showed a slight increase of 11.1%. The product per substrate (Y$_{P/S}$) was at 0.082 g/g, and product per biomass (Y$_{P/X}$) was at 0.5 g/g. However, surfactin yield and DCW (1.6 g/L, 6.68 g/L, respectively) decreased sharply under high aeration volume (0.13 vvm), where the values were 58.1% and 18.9% lower than those for the control

(3.82 g/L, 8.24 g/L). The yield of surfactin (2.55 g/L) and DCW (6.1 g/L) decreased by 33.2% and 26%, respectively, at the lower aeration volume (0.02 vvm).

As noted above, pH is a key factor affecting the synthesis of surfactin by LJ-31. Hence, the pH was optimized under the conditions of C/N 11.68/1 and aeration volume of 0.03 vvm. The results shown in Table 3 indicate that surfactin synthesis was possible under weak acidic (pH 6.5) conditions. Surfactin synthesis was severely inhibited at a pH below or above 6.5. When pH was maintained at 6, the yield of surfactin decreased sharply by 37.6% compared with that at pH 6.5 (4.1 g/L), and when pH was maintained at 7–7.5, surfactin yield slightly decreased by 6–10%. Therefore, pH has a significant effect on surfactin synthesis, and this finding is similar to the conclusions reported by Cosby et al. [27] and Yi [29] et al.

When all the results are taken together, LJ-31 can effectively synthesize surfactin under the following conditions: C/N ratio of 11.68/1, aeration volume of 0.03 vvm, and pH of 6.5. At these conditions, changes in cell growth, surfactin yield, and kinetic parameters are shown in Figure 8. The product per substrate ($Y_{P/S}$) at 22 h was 0.082 g/g, an increase of 49%, while product per biomass ($Y_{P/X}$) was 0.55 g/g, an increase of 57.1% (Figure 8A). At 10 h, the maximum specific production rate ($q_{pmax}$) for surfactin was 0.2 h$^{-1}$. The maximum specific growth rate at 5 h was 0.38 h$^{-1}$ (Figure 8B) and increased by 35.7%. The surfactin yield reached 27.5 mmol/mol sucrose (15.8% of the theoretical yield). Interestingly, when the pH is maintained at its initial value (pH 6.5), surfactin synthesis was most amenable and the specific growth rate only changed after optimization of the C/N ratio and dissolved oxygen. Therefore, regulation and optimization of the first two conditions (C/N and dissolved oxygen) affected the specific growth rate of the strain, indicating that when the strain is provided with an appropriate growth environment, this accelerates accumulation of the strain (DCW) ahead of initiating protein expression, thus improving strain productivity and increasing the surfactin yield. We suggest that pH affects strain characteristics in terms of surfactin production. When the pH was maintained at 6.5 to eliminate the effect of ammonium chloride, the yield of surfactin was significantly improved (Figure 7A). At the same time, the results provide some ideas for improving the yield of surfactin in the later period of fermentation; for example, modifying the anabolic pathways of the strain, such as the precursor supply module, transcription module, and repressor competition pathway, could improve the yield. Thus, a combination of ALE and genetic engineering techniques can obtain strains with performance advantages.

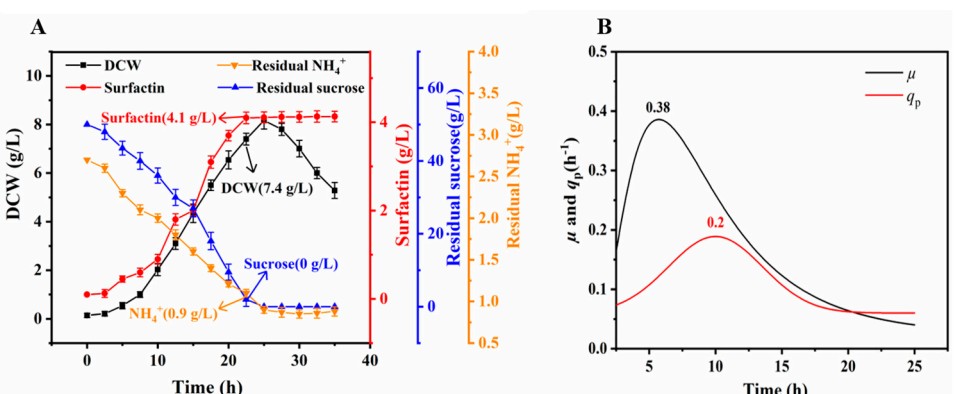

**Figure 8.** The fermentation process and related kinetic parameters ($\mu$, $q_p$) in an optimized fermentation system. (**A**) Changes in substrate, DCW, and surfactin yield. (**B**) The specific yield of surfactin ($q_p$) and specific growth rate ($\mu$).

As shown in Table 4, recombinant strain LJ-31 has a significant advantage compared with most wild-type and commercial bacteria that can use inorganic nitrogen in terms of production or growth performance, while also allowing for gene editing. Overall, *B. subtilis* 168, a lab strain with a preference for organic nitrogen, was successfully transformed into a chassis strain with efficient utilization of NH$_4$Cl and able to effectively produce surfactin.

**Table 4.** Synthesis of surfactin by different strains using inorganic nitrogen sources.

| Process, Strains | Nitrogen Source | Biomass | $P_{max}$ | Reference |
|---|---|---|---|---|
| LJ-31 (*B. subtilis* 168 derivative bacteria) batch fermentation in bioreactor | $NH_4Cl$ | 7.4 g/L (DCW) | 4.1 (g/L) | This study |
| *B. subtilis* ATCC 21332 (commercial strain) batch fermentation in shake flasks | $NaNO_3$, $KNO_3$ | 10.2 ($OD_{600}$) | 3.054 (g/L) | [22] |
| *Bacillus subtilis* SPB1 strain wild-type batch fermentation in bioreactor | Urea, $NH_4Cl$, kerosene | $20 \times 10^8$ (cells/mL) | 4.922 (g/L) | [30] |
| *Bacillus velezensis* H3 (wild-type) batch fermentation in shake flasks | $(NH_4)_2SO_4$ | — | 0.4–0.5 (g/L) | [31] |
| *B. cereus* BCS0 (wild-type) batch fermentation in shake flasks | Urea, $NH_4Cl$, $NaNO_3$ | — | 0.7–1.22 (g/L) | [32] |
| *B. cereus* BCS1 (wild-type) batch fermentation in shake flasks | Urea, $NH_4Cl$, $NaNO_3$ | — | 0.11–0.5 (g/L) | [32] |
| *B. cereus* BCS2 (wild-type) batch fermentation in shake flasks | Urea, $NH_4Cl$, $NaNO_3$ | — | 0.51–2.71 (g/L) | [32] |
| *B. cereus* BCS3 (wild-type) batch fermentation in shake flasks | Urea, $NH_4Cl$, $NaNO_3$ | — | 1.7–2.91 (g/L) | [32] |
| 168 (JABs24) and 3NA (JABs32) batch fermentation in bioreactor | $(NH_4)_2SO_4$ | — | 2.56–2.68 (g/L) | [15] |

(—) Not mentioned in the article.

### 3.5. Composition of Surfactin Synthesized by LJ-31

Surfactin components produced from LJ-31 were analyzed by HPLC. As shown in Figure 9, the main components of surfactin synthesized by recombinant strain LJ-31 were the same as the standard surfactin (CAS:24730-31-2; Sigma Company), but the proportion of each component was significantly different. Peaks labeled with a, b, c, and d correspond to (a) $C_{13}$-surfactin, (b) iso-$C_{14}$-surfactin, (c) n$C_{14}$-surfactin, and (d) anteiso-$C_{15}$-surfactin as identified in our previous work [10,33]. The proportion of surfactin isoforms produced by strain LJ-31 in the synthetic medium was $C_{13}$-surfactin 13.3% $\pm$ 1.5, $C_{14}$-surfactin 44.02% $\pm$ 1.55, and $C_{15}$-surfactin 32.79% $\pm$ 1.6. Our previous work also showed that surfactin with a higher proportion of $C_{14}$-surfactin had better surface activity, higher oil washing efficiency and better emulsification activity, and improved the water wettability of solid surfaces [10]. Therefore, the recombinant strain LJ-31 produces surfactin at a low cost, and in addition, the synthesized surfactin performs well in its proposed applications.

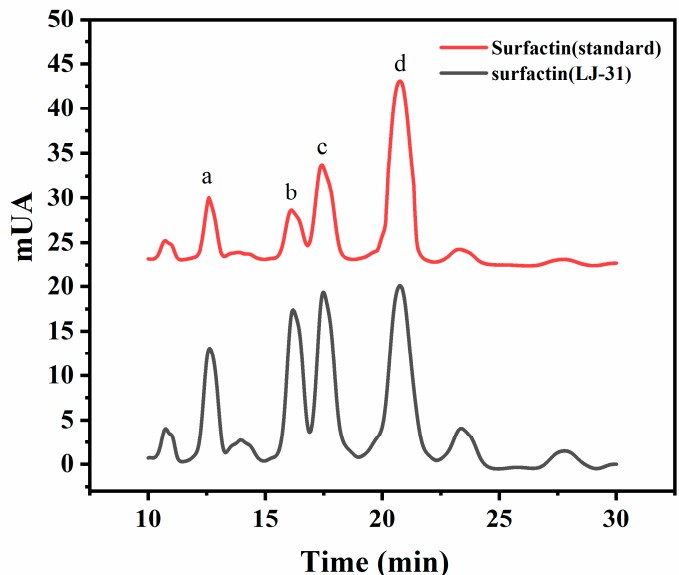

**Figure 9.** Composition analysis of surfactin products: a-$C_{13}$-surfactin, b-iso-$C_{14}$-surfactin, c-n$C_{14}$-surfactin, and d-anteiso-$C_{15}$-surfactin.

## 4. Conclusions

The adaptive laboratory evolution (ALE) can be used to improve *B. subtilis* 168 growth using NH$_4$Cl as the sole nitrogen source. The variant LJ-3 isolated by ALE is a promising engineering chassis that can efficiently use NH$_4$Cl as the sole nitrogen source. Reintroduction of the *sfp* gene into strain LJ-3 led to restoring its ability to synthesize surfactin, and the highest surfactin titer reached 4.1 g/L after 22 h fermentation. Our study contributes to producing lower cost surfactin.

**Author Contributions:** Conceptualization, J.L. and S.L.; methodology, J.L., W.T. and S.L.; formal analysis, J.L., S.Y. and S.L.; investigation, data curation, J.L., W.T., Z.Y. and S.L.; resources, S.Y. and Z.Y.; writing—original draft preparation, J.L. and S.L.; writing—review and editing J.L., W.T., S.L., S.Y. and Z.Y.; supervision, W.T., S.L., S.Y. and Z.Y. All authors have read and agreed to the published version of the manuscript.

**Funding:** This work was supported by the National Key Research and Development Program of China (grant no. 2022YFC2105202) and the Jiangsu Synergetic Innovation Center for Advanced Bio-Manufacture.

**Institutional Review Board Statement:** Not applicable.

**Informed Consent Statement:** Not applicable.

**Data Availability Statement:** The research data are not shared.

**Conflicts of Interest:** The authors declare no conflict of interest.

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
