# Peer review of "Adaptive Laboratory Evolution of Bacillus subtilis 168 for Efficient Production of Surfactin Using NH4Cl as a Nitrogen Source"

_fermentation, doi:10.3390/fermentation9060525_

Round 1
Reviewer 1 Report
The authors described adaptive laboratory evolution techniques for Bacillus subtilis 168 producing surfactin. The authors improved growth rate, production of surfactin of B. subtilis 168 variant LJ-31 using only NH4Cl as a nitrogen source. This genetically developed strain LJ-31 will be applied for academic and industrial field broadly. However, there are minor points to improve this manuscript. Here is list of my comments.
1. Every written units including h, C, L, mL, etc need a space between number. For example, Page 5 line 139, "incubated for 12h at.." need to be corrected as "incubated for 12 h at..". Please correct all space.
2. Page 12 Figure 8. CDW should be corrected to DCW.
Author Response
Thank you very much for your professional suggestion, and your advice helps to improve our paper. Every written units of article including h, C, L, mL, etc., has Spaces added between numbers. Page 10 Figure 7 and page 12 Figure 8 CDW has been corrected to DCW.

Reviewer 2 Report
There are no fundamental remarks to this work, but Figure 1 needs to be improved. In the figure there are signs BS37 and LJ3, but in the caption only LJ-3 is described. The labels NaOH, HCl and antifoam are not obvious, these are not fermentation optimization parameters. It is necessary to redraw a figure and make signs more specific and clear.
Figure 6. On the Plot "B" it is desirable to show surfactin production by the LJ-3 strain as a negative control.
Author Response
The description of B. velezensis BS-37 in Figure 1 (c) is supplementary described and the figure is redrawn with more specific and clear symbols. In Figure 6, we added the surface hormone yield of LJ-3 strain to the negative control.
